# Non-invasive fibrosis algorithms are clinically useful for excluding cirrhosis in prisoners living with hepatitis C

Timothy Papaluca[1], Anne Craigie[1], Lucy McDonald[1], Amy Edwards[1], Michael MacIsaac[1], Jacinta A. Holmes[1], Matthew Jarman[1], Tanya Lee[1], Hannah Huang[1], Andrew Chan[1], Mark Lai[1], Vijaya Sundararajan[2,3], Joseph S. Doyle[4,5,6], Margaret Hellard[4,5,6], Mark Stoove[4,5], Jessica Howell[1,4,5], Paul Desmond[1], David Iser[1], Alexander J. Thompson[1] *

1 Department of Gastroenterology, St Vincent's Hospital and the University of Melbourne, Melbourne, Australia, 2 Department of Medicine, Dentistry and Health Sciences, University of Melbourne, Melbourne, Australia, 3 Department of Public Health, La Trobe University, Melbourne, Australia, 4 Burnet Institute, Melbourne, Australia, 5 Department of Epidemiology and Preventative Medicine, Monash University, Melbourne, Australia, 6 Department of Infectious Diseases, The Alfred and Monash University, Melbourne, Australia

* alexander.THOMPSON@svha.org.au

**Data Availability Statement:** The data included in this analysis includes sensitive information regarding prisoners under the care of the Department of Justice and Regulation, Victorian

## Abstract

### Background and aims

Prison-based HCV treatment rates remain low due to multiple barriers, including accessing transient elastography for cirrhosis determination. The AST-to-platelet ratio index (APRI) and FIB-4 scores have excellent negative predictive value (NPV) in hospital cohorts to exclude cirrhosis. We investigated their performance in a large cohort of prisoners with HCV infection.

### Methods

This was a retrospective cohort study of participants assessed by a prison-based hepatitis program. The sensitivity, specificity, NPV and positive predictive value (PPV) of APRI and FIB-4 for cirrhosis were then analysed, with transient elastography as the reference standard. The utility of age thresholds as a trigger for transient elastography was also explored.

### Results

Data from 1007 prisoners were included. The median age was 41, 89% were male, and 12% had cirrhosis. An APRI cut-off of 1.0 and FIB-4 cut-off of 1.45 had NPVs for cirrhosis of 96.1% and 96.6%, respectively, and if used to triage prisoners for transient elastography, could reduce the need for this investigation by 71%. The PPVs of APRI and FIB-4 for cirrhosis at these cut-offs were low. Age ≤35 years alone had a NPV for cirrhosis of 96.5%. In those >35 years, the APRI cut-off of 1.0 alone had a high NPV >95%.

Government. As such, in keeping with Department of Justice regulations, this data cannot be made publicly available. Data queries regarding the hepatitis C non-invasive fibrosis algorithm dataset can be forwarded to Bernadette De Graff, Clinical Research Compliance Manager, Department of Gastroenterology, St Vincent's Hospital Melbourne, Bernadette.degraff@svha.org.au.

**Funding:** TP received funding from an Australian Government Research Training Program Scholarship and the Department of Gastroenterology, St Vincent's Hospital Melbourne; https://urldefense.com/v3/__https://www.education.gov.au/research-training-program__;!!LUsMDrd6!yF7XCdsDwModOT7ENzQHtsvv3lAco8amgz6Bf0hZtRF8uM2Os3mbmU1by4H_6ApxdVSrzNSo$ AJT and MH received funding from the National Health and Medical Research Council of Australia (NHMRC) Practitioner Fellowships 1142976 and 1112297. https://urldefense.com/v3/__https://www.nhmrc.gov.au/__;!!LUsMDrd6!yF7XCdsDwModOT7ENzQHtsvv3lAco8amgz6Bf0hZtRF8uM2Os3mbmU1by4H_6ApxdT5HBwwc$ JH is funded by a University of Melbourne CR Roper Fellowship and a NHMRC program grant. https://urldefense.com/v3/__https://staff.unimelb.edu.au/mdhs/research-development/research-collaboration-and-funding/faculty-trust-fellowships/cr-roper-fellowship__;!!LUsMDrd6!yF7XCdsDwModOT7ENzQHtsvv3lAco8amgz6Bf0hZtRF8uM2Os3mbmU1by4H_6Apxdbmg15Ep$ This work was supported by NHMRC Program grant 1132902 and Partnership grant 1116161. https://urldefense.com/v3/__https://www.nhmrc.gov.au/__;!!LUsMDrd6!yF7XCdsDwModOT7ENzQHtsvv3lAco8amgz6Bf0hZtRF8uM2Os3mbmU1by4H_6ApxdT5HBwwc$ The funders had no role in study design, data collection and analysis, decision to publish, or preparation of the manuscript.

**Competing interests:** The authors have no disclosures in relation to this work.

## Conclusion

APRI and FIB-4 scores can reliably exclude cirrhosis in prisoners and reduce requirement for transient elastography. This finding will simplify the cascade of care for prisoners living with hepatitis C.

## Introduction

The advent of direct acting antivirals (DAAs) has revolutionised the treatment of chronic hepatitis C virus (HCV) infection. People living with HCV can now be treated with all oral, short duration, and well tolerated DAA regimens which achieve cure rates in excess of 95% [1,2]. In this context, the World Health Organization (WHO) has established targets for HCV elimination, including an 80% reduction in incident infections and a 65% reduction in HCV related mortality by 2030 [3]. Mathematical modelling has demonstrated that to achieve these targets, concerted efforts must be made to treat key populations with high risk for transmission including people who inject drugs (PWID) [4].

Prisons provide an excellent opportunity to identify and treat people living with HCV given the over-representation of people with injecting drug use histories and the associated high prevalence of chronic HCV [5]; HCV seroprevalence exceeds 15% in prisons globally, and in Australia is as high as 50% amongst incarcerated PWIDs [6,7]. Multiple data have now demonstrated that prison-based HCV treatment is safe and effective and can reach prisoners in large numbers [8–10]. Despite this, rates of HCV treatment in prisons worldwide remain low due to multiple barriers [11]. These include frequent prisoner transfer interrupting treatment, short sentence durations, the need for hospital transfer for specialist review and limited resourcing for HCV testing and treatment [12,13]. One important barrier is access to TE (e.g. FibroScan®) for cirrhosis determination in the correctional setting [14].

Cirrhosis determination prior to HCV DAA initiation is important for decision-making regarding treatment regimen and duration, and to identify those who require ongoing specialist referral for cirrhosis management and hepatocellular cancer (HCC) surveillance [15]. TE is the most accurate non-invasive tool for cirrhosis determination and has largely replaced liver biopsy in the community [16]. Access to TE within the correctional sector however is challenging and remains a significant barrier to care due to its high cost, limited availability of equipment within prison health services, frequent need for prisoner transfer and issues with security [10,17].

Based on routine blood tests, the AST-to-platelet ratio index (APRI) and the FIB-4 score were developed to predict advanced fibrosis and cirrhosis, using community recruited samples [18–21]. Both scores have subsequently been validated and demonstrate excellent negative predictive value for excluding advanced fibrosis or cirrhosis. Indeed, the WHO recommends both APRI and FIB-4 as cost-effective tools for the assessment of hepatic fibrosis in resource-limited settings [22]. Prisoner populations, however, are characterised by male gender, younger age, as well as multiple high-risk behaviours that differ considerably from the hospital populations used to validate these algorithms. Validation of simple serum-based fibrosis markers for use in the correctional setting to triage prisoners at risk of cirrhosis would be clinically useful and help improve the cascade of care by reducing treatment costs and the time delay and logistical challenges associated with TE.

We have therefore evaluated the accuracy of non-invasive fibrosis algorithm scores including APRI and FIB-4 as a practical triage tool for excluding cirrhosis in the correctional setting. This analysis was conducted using a large cohort of prisoners with HCV infection with a view

to simplifying clinical pathways by reducing the number of prisoners requiring TE, therefore improving the HCV cascade of care.

## Materials and methods

### Participants

This was a retrospective cohort study investigating the performance of the APRI and FIB-4 in a prison setting. The cohort provided prospectively collected data from all prisoners with HCV infection evaluated through the Statewide Hepatitis Program in Victoria, Australia. Prisoners evaluated for HCV infection were consecutively recruited from November 2015. This model of care has been described in detail previously [8]. Prisoners who self-reported a prior HCV diagnosis or who were identified as HCV seropositive whilst incarcerated were referred to the Statewide Hepatitis Program for face-to-face protocol-driven assessments with a program nurse. Clinical assessments were conducted at all 16 adult prisons throughout Victoria, which are serviced by the program. Pathology testing was performed by the diagnostic laboratory affiliated with each correctional facility. A database was created containing participants' baseline clinical characteristics, blood test results and liver stiffness measurement (LSM) median scores, interquartile ranges (IQR) and IQR:median ratios. Blood-based investigations that were recorded included haemoglobin, white blood cell count, platelet count, creatinine, urea, sodium, alanine aminotransferase (ALT), aspartate aminotransferase (AST), gamma-glutamyl transpeptidase (GGT), alkaline phosphatase (ALP), bilirubin, albumin, international normalised ratio (INR), hepatitis B/C serology, HIV serology, and HCV genotype and viral load.

Prisoners were excluded if they had insufficient blood test results to calculate the APRI or FIB-4 scores within 6 months of TE, if TE had not been performed, if IQR:median ratio exceeded 0.3 or the absence of current HCV infection, defined as detectable HCV RNA (Fig 1).

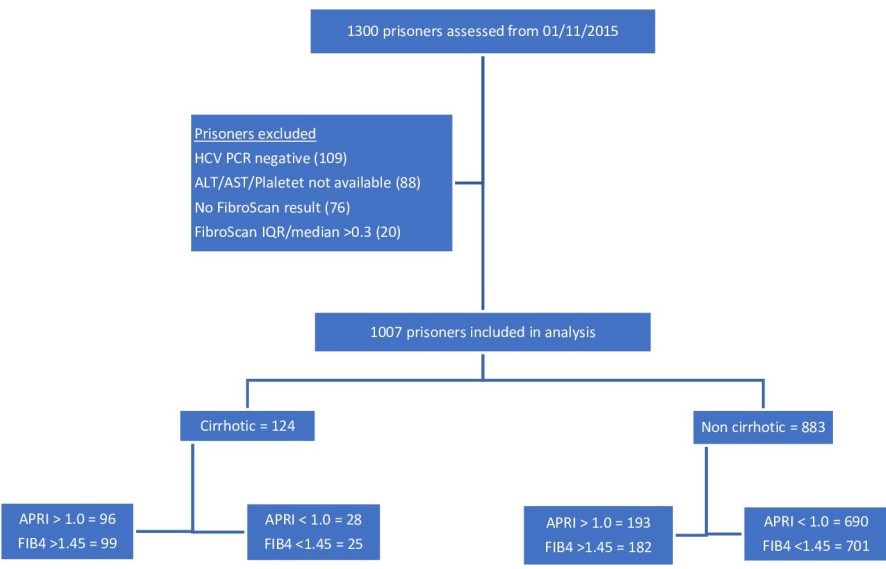

**Fig 1. Consort diagram.** Legend IQR, interquartile range.

## Liver cirrhosis determination

TE was used for cirrhosis determination at each prisoner's initial clinical assessment to guide appropriate management. Cirrhosis was defined as a median LSM of ≥12.5kPa, in keeping with Australian HCV consensus guidelines [23,24]. Blood-based and radiologically investigations were also arranged at this review, where appropriate. TE was performed using FibroScan® (Echosens) with the 'M' or 'XL' probe, as determined by the participant's body mass index (BMI) and body habitus. Measurements were collected in the mid-axillary line until ten valid readings were recorded. The LSM results was deemed valid only when the IQR/median ratio was less than 0.3 and when performed prior to HCV DAA initiation. All transient elastography was performed by three hepatology clinical nurse consultants who had performed at least 100 supervised examinations for credentialing. The APRI and FIB-4 scores were then determined for each prisoner. The APRI was calculated as $[(AST/Upper\ limit\ of\ normal\ of\ AST)\ x\ 100]/[Platelets(10^9/L)]$ [18]. The FIB-4 score was calculated as $[(age\ x\ AST)/(platelets(10^9/L)\ x\ ALT^{1/2})]$ [19].

## Statistical methodology and analysis

Statistical analyses were performed using STATA 12.0 (StataCorp LP, College Station, TX, USA). Categorical data were described as number and percentage and interval data as median and IQR.

Data were used to determine the sensitivity, specificity, PPV and NPV of the APRI and FIB-4 scores to diagnose or exclude cirrhosis using standard analytical approaches. Performance of APRI and FIB-4 were assessed against gold-standard TE assessments. Cut-offs of 1.0 and 2.0 for APRI, and 1.45 and 3.25 for FIB-4 were selected on the basis of having been previously validated in other data for the diagnosis of cirrhosis and advanced fibrosis (METAVIR F3/4), respectively [18–20]. Two analyses for FIB-4 at these thresholds were considered, including for advanced fibrosis (LSM ≥9.5kPa) (for which this algorithm was originally validated) and for cirrhosis (≥12.5kPa). The reduction in requirement for TE was also determined should prisoners be triaged for TE using these algorithms cut-offs.

In addition to APRI and FIB-4, given the young average age of prisoner populations globally and the slow natural history of HCV infection [25], the sensitivity, specificity, NPV and PPV of binary age thresholds alone to predict or exclude cirrhosis were also analysed, to determine if stratifying fibrosis assessment by age could minimise the need for TE while maintaining high NPV. The age thresholds were determined by the data and the quartile values (25th, 50th and 75th) were selected for analysis. The cohort was then separated by these age values and we determined the predictive value of APRI at a cut off of 1.0 in these groups separately to contribute to new clinical pathways for fibrosis assessment for the entire prison cohort incorporating age categories. This novel fibrosis assessment pathway was then validated using data from an additional cohort of prisoners.

## Ethics statement

This study was approved by the St Vincent's Hospital Human Research Ethics Committee. The study was granted a waiver of consent as all data were analysed anonymously. The study conformed to the ethical guidelines of the 1975 Declaration of Helsinki, Good Clinical Practice Guidelines, and regulatory requirements.

## Results

### Prisoner characteristics

In total, data from 1300 was assessed. Two hundred and ninety three prisoners were excluded including 109 who were HCV PCR negative, 88 who had insufficient blood test results to

calculate APRI and/or FIB-4 (n = 88), 76 who did not undergo TE (n = 76) and 20 who had LSM IQR:median >0.3 (n = 20) (Fig 1). As such, 1007 were included in the analysis. Baseline characteristics are described in Table 1. In short, prisoners were prominently male (89%), the median age was 41 [35–47] years, HCV GT3 infection was the most common genotype (46%), and there were low rates of HBV and HIV coinfection (1%). One hundred and twenty-four individuals (12%) were cirrhotic, as defined by TE (LSM ≥ 12.5kPa, n = 124/1007). The median APRI score was 0.66 [0.41–1.09] and median FIB-4 score was 1.02 [0.69–1.54] (Table 2). The median time between the collection of blood-based investigations and transient elastography was 34 days [IQR 13–86 days].

## Performance of APRI or FIB-4 scores for cirrhosis in prisoners

The performance of the APRI and FIB-4 scores for cirrhosis (LSM ≥ 12.5kPa) at previously validated thresholds are presented in Table 3. An APRI score cut-off of 1.0 had a sensitivity and specificity for cirrhosis of 77.4% and 78.1%, respectively. The NPV of APRI <1.0 to exclude cirrhosis was 96.1%. If this cut-off was incorporated into prison fibrosis assessment protocols where only prisoners with an APRI ≥1.0 were referred for TE, the need for FibroScan® would reduce by 71% (n = 718/1007). The PPV at this threshold was low (33.2%), meaning that these prisoners would still require TE for cirrhosis confirmation. At the higher APRI cut-off of 2.0, the PPV was higher (48.4%), but resulted in lower NPV (91.4%) and sensitivity (36.3%)–using an APRI cut-off of 2.0 alone to exclude cirrhosis would have meant that 64% (n = 79/124) cases of cirrhosis were missed (S1 Table).

Similar patterns were observed using different FIB-4 cut-offs, with the lower cut-off score of 1.45 associated with a high NPV for the exclusion of cirrhosis, and a low PPV (Table 3). At a higher cut-off score of 3.25, the specificity of the score was enhanced, however the sensitivity and NPV were significantly reduced (Table 3).

**Table 1. Prisoner baseline characteristics.**

| Baseline characteristics | n = 1007 |
|---|---|
| Age, median [IQR] | 41 [35–47] |
| BMI, median [IQR] | 28.4 [25.3–32.2] |
| Male sex, n (%) | 899 (89) |
| Indigenous Australian, n (%) | 133 (13) |
| HCV viral load IU/mL, median [IQR] | 757,000 [193,250–3,130,750] |
| HCV Genotype, n, (%) | |
| • GT1a | 435 (43) |
| • GT1b | 33 (3) |
| • GT2 | 26 (3) |
| • GT3 | 466 (46) |
| • GT4 | 2 (0.5) |
| • GT6 | 8 (0.5) |
| • NA | 37 (4) |
| HBsAg positive, n (%) | 14 (1) |
| HIV Ab positive, n (%) | 12 (1) |
| ALT, median [IQR] | 81 [53–132] |
| AST, median [IQR] | 52 [36–79] |
| Platelets, median [IQR] | 234 [192–276] |
| Cirrhotic, n (%) | 124 (12) |

Baseline characteristics of prisoner population at initial assessment. Legend, BMI, body mass index, GT, genotype.

**Table 2. Results of cirrhosis determination using APRI, FIB-4 and elastography.**

| Cirrhosis determination | n = 1007 |
|---|---|
| APRI score, n (%) | |
| • < 1.0 | 718 (71) |
| • 1.01–2.0 | 196 (19) |
| • > 2.0 | 93 (9) |
| FIB-4 score, n (%) | |
| • < 1.45 | 726 (72) |
| • 1.45–3.25 | 236 (23) |
| • >3.25 | 45 (45) |
| Liver Stiffness Measurements, n (%) | |
| • < 6 kPa | 356 (36) |
| • 6–9.4 kPa | 435 (43) |
| • 9.5–12.4 kPa | 92 (9) |
| • ≥ 12.5 kPa | 124 (12) |

Results of cirrhosis determination using APRI, FIB-4 and elastography.

The performance of FIB-4 for diagnosing or excluding advanced fibrosis (LSM ≥9.5kPa) at previously validated threshold was also analysed (S1 Table).

## Age thresholds for the exclusion of cirrhosis

The sensitivity, specificity, NPV and PPV of age thresholds alone for cirrhosis were determined. The binary age thresholds selected for analysis were 35, 41 and 47 years, representing the quartile values for the cohort. The binary variable of prisoner age ≤35 achieved a NPV for cirrhosis of 96.5%, due to the low prevalence in this group (3.5%, n = 9/258) (Table 4). If no formal fibrosis assessment was performed for prisoners aged ≤35, nine cases of cirrhosis would be missed, representing 1% of the total cohort. The NPV of the age thresholds 41 and 47 years alone to exclude cirrhosis decreased to 94.7% and 91.6%, respectively, due to the increasing prevalence of cirrhosis in these age groups (Table 4). Furthermore, as the age threshold

**Table 3. Accuracy of APRI and FIB4 in predicting cirrhosis as compared to transient elastography.**

| | | Liver stiffness measurement | | Sensitivity % | Specificity % | PPV % | NPV % |
|---|---|---|---|---|---|---|---|
| | All prisoners (n = 1007) n (%) | < 12.5kPa (n = 883) n (%) | ≥ 12.5kPa (n = 124) n (%) | | | | |
| For prediction of cirrhosis | | | | | | | |
| APRI | | | | | | | |
| ≤ 1.0 | 718 (71) | 690 (78) | 28 (23) | 77 | 78 | 33 | 96 |
| > 1.0 | 289 (29) | 193 (22) | 96 (77) | | | | |
| ≤ 2.0 | 914 (91) | 835 (95) | 79 (64) | 36 | 95 | 48 | 91 |
| > 2.0 | 93 (9) | 48 (5) | 45 (36) | | | | |
| FIB4 | | | | | | | |
| ≤ 1.45 | 713 (71) | 688 (78) | 25 (20) | 80 | 78 | 34 | 97 |
| > 1.45 | 294 (29) | 195 (22) | 99 (80) | | | | |
| ≤ 3.25 | 960 (95) | 871 (99) | 89 (72) | 28 | 99 | 75 | 91 |
| > 3.25 | 47 (5) | 12 (1) | 35 (28) | | | | |

Sensitivity, specificity, NPV and PPV of the APRI and FIB-4 scores to diagnose or exclude cirrhosis (LSM≥12.5kPa) represented as percentages. Legend, NPV, negative predictive value, PPV, positive predictive value.

**Table 4. Performance of age categories for predicting cirrhosis as compared to transient elastography.**

| | All prisoners (n = 1007) n (%) | Liver stiffness measurement | | Sensitivity % | Specificity % | PPV % | NPV % |
| | | < 12.5kPa (n = 883) n (%) | ≥ 12.5kPa (n = 124) n (%) | | | | |
|---|---|---|---|---|---|---|---|
| For prediction of cirrhosis | | | | | | | |
| Age | | | | | | | |
| ≤ 35 | 258 (26) | 249 (28) | 9 (7) | 92 | 28 | 26 | 97 |
| > 35 | 749 (74) | 634 (72) | 115 (93) | | | | |
| ≤ 41 | 528 (52) | 500 (57) | 28 (23) | 77 | 57 | 20 | 95 |
| > 41 | 479 (48) | 383 (43) | 96 (77) | | | | |
| ≤ 47 | 752 (75) | 689 (78) | 63 (51) | 49 | 78 | 24 | 92 |
| > 47 | 255 (25) | 194 (22) | 61 (49) | | | | |

Ability of age alone to diagnose or exclude cirrhosis, with TE as the reference standard. Legend, NPV, negative predictive value, PPV, positive predictive value.

increased, the number of prisoner with cirrhosis who were missed, increased. We also calculated the NPV using the combination of age and APRI (APRI cut-off of 1.0 for prisoners aged ≤35 and >35 to exclude cirrhosis (Table 4)). Amongst the age ≤35 years subgroup, an APRI cut-off of 1.0 was associated with a NPV of 99.0%, however would result in 26% of this age group requiring TE who had an APRI >1.0. In those >35 years old, the NPV of APRI 1.0 remained satisfactory and was >95%, which could reduce the need for TE by 70% in this age group (Table 5).

Finally, a novel fibrosis assessment algorithm of performing TE only in prisoners >35 years and only when APRI ≥1.0 would achieve a NPV of >95% for cirrhosis and reduce the need for TE by 78% of the entire cohort (Fig 2). The NPV for cirrhosis was sustained at 97% when this fibrosis assessment pathway was applied to a validation cohort of 189 prisoners, 11% (n = 20/189) of whom were cirrhotic, and referral for TE would be reduced by 83% (n = 156/189) (S2 Table).

## Discussion

HCV treatment within prisons is important for global viral elimination, however the requirement for TE for fibrosis assessment presents barriers to efficient HCV care in custodial settings. Our data demonstrate that the APRI and FIB-4 serum scores can be useful to stratify risk of cirrhosis in a prison population, using validated thresholds from hospital-based cohorts.

**Table 5. Performance of APRI in different age categories for predicting cirrhosis as compared to transient elastography.**

| | All prisoners (n = 1007) n (%) | Liver stiffness measurement | | Sensitivity % | Specificity % | PPV % | NPV % |
| | | < 12.5kPa (n = 883) n (%) | ≥ 12.5kPa (n = 124) n (%) | | | | |
|---|---|---|---|---|---|---|---|
| For prediction of cirrhosis | | | | | | | |
| Age + APRI | | | | | | | |
| ≤ 35 | | | | | | | |
| APRI ≤ 1.0 | 192 (19) | 190 (21) | 2 (2) | 78 | 76 | 11 | 99 |
| APRI > 1.0 | 66 (7) | 59 (7) | 7 (6) | | | | |
| > 35 | | | | | | | |
| APRI ≤ 1.0 | 526 (52) | 500 (57) | 26 (21) | 78 | 79 | 40 | 95 |
| APRI > 1.0 | 223 (22) | 134 (15) | 89 (71) | | | | |

The performance of APRI in age categories ≤ 35 and >35 to diagnose or exclude cirrhosis. Legend, NPV, negative predictive value, PPV, positive predictive value.

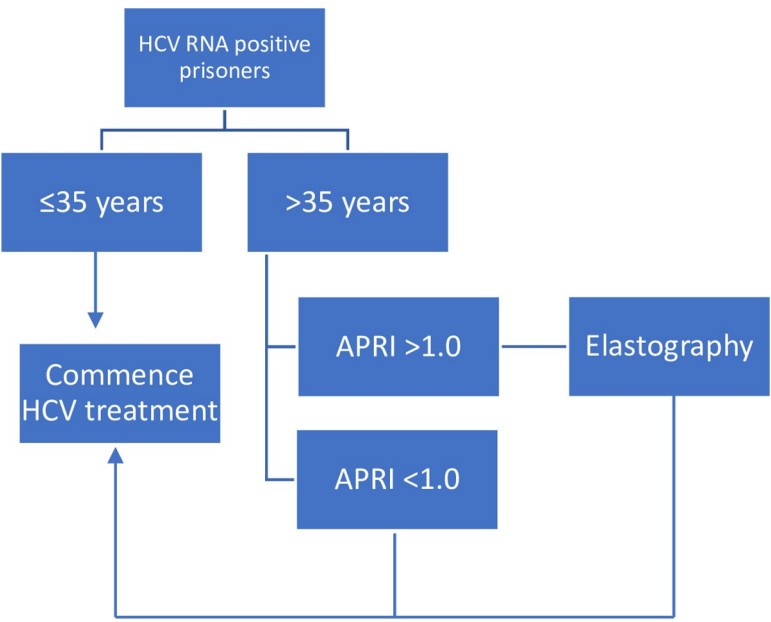

**Fig 2. A novel clinical pathway for fibrosis assessment prior to HCV treatment in prisons.** This pathway achieves a NPV for cirrhosis of >95% in all prisoners and minimises the need for FibroScan by 78%.

When applied to our cohort of prisoners with HCV infection, APRI and FIB-4 thresholds of 1.0 and 1.45, respectively, were associated with NPVs for cirrhosis in excess of 96%. The sensitivity and PPV for both algorithms, however, was low. The data, therefore, demonstrate that the strength of these algorithms lies in the exclusion, rather than confirmation, of cirrhosis. The NPV of binary age thresholds for excluding cirrhosis were also investigated. The NPV of age ≤35 alone exceeded 96%, and could be incorporated into fibrosis assessment protocols to improve HCV treatment throughput. The use of APRI and FIB-4 have been prioritised by the WHO for cirrhosis determination in low- and middle-income countries due to their availability and low cost [18]. Similarly, prison health services provide complex care to prisoners in large numbers, including for HCV infection, yet operate under significant fiscal restrictions. Therefore, the use of these indices to minimise cost and streamline care should be expanded.

Both APRI (threshold of 1.0) and FIB-4 (threshold 1.45) had NPVs of >96% for cirrhosis. We believe that this NPV is such that a second test for fibrosis determination is not required. The use of these thresholds to determine need for TE would result in a 70% reduction in the number of prisoners being referred for TE. As such, these indices could help simplify pathways of care for HCV in prisons and promote increased treatment capacity. While mobile FibroScan® units are available, high equipment costs and mandatory training requirements mean that it is not feasible to have them routinely available in many prisons. As there are significant costs and logistics associated with transporting prisoners to tertiary hospitals or a central prison where a FibroScan® might be available, the reliance on TE for fibrosis assessment represents a significant challenge for expanding HCV treatment coverage in prisons. Using these algorithms to reduce the need for TE in the majority while maintaining a high NPV is significant, as delays in facilitating this investigation are at odds with the short average duration of incarceration, meaning some prisoners would be released without treatment. The NPV of APRI and FIB-4 are influenced by cirrhosis prevalence, and our cohort of predominately young male prisoners is well suited for their use.

Our data also demonstrate that age ≤35 years has a very high NPV for cirrhosis (96.5%), and that there is little clinical utility in applying blood-based algorithms or TE to this population. Evaluation of liver fibrosis in those ≤35 years might be restricted to people with clinical risk factors for cirrhosis, such as multiple spider naevi. In prisoners >35 years there is a higher prevalence of cirrhosis, and in this group formal fibrosis evaluation is important. APRI <1.0 had a NPV for cirrhosis of >95% in prisoners >35 years, reducing the need for TE by 70% in this age group. Therefore, LSM could be reserved for those prisoners >35 years who have APRI >1.0. Combined, this approach achieves a NPV of >95% across the entire cohort and minimised the need for TE by 78% (Fig 2). When applied to a validation cohort, the NPV of this approach was sustained and <20% would require referral for TE.

Despite the high NPV of the FIB4, APRI and age ≤ 35 to exclude cirrhosis, a small number of cirrhotic prisoners were miscategorised. Using a FIB-4 cut off of 1.45 for TE, 2% (n = 25/1007) of the overall cohort and 20% (n = 25/124) of cirrhotics were incorrectly identified as non-cirrhotic. This was similar for an APRI threshold of 1.0 (3% miscategorised, n = 28/1007). All miscategorised prisoners were Child-Turcotte-Pugh class A and had preserved platelet counts. As such, the main implication for those miscategorised is exclusion from HCC surveillance. Achieving cure for these prisoners however will lead to significant reductions in their HCC risk and more broadly, non-HCC liver related morbidity and mortality [26]. Therefore, the reduction in prison-based HCV treatment throughput if TE is required for all prisoners must be considered. By stratifying the need for TE using fibrosis algorithms, it is anticipated that program efficiencies would be streamlined and that a greater number of prisoners would be treated, preventing the progression to cirrhosis and HCC in many.

The data demonstrate that the strength of APRI and FIB-4 are for excluding, rather than diagnosing cirrhosis. The higher APRI and FIB4 scores at thresholds of 2.0 and 3.25, which are recommended in WHO guidelines, were analysed for their ability to diagnose cirrhosis. Whilst achieving high specificity at these thresholds (>95% for both APRI 2.0 and FIB4 3.25), their sensitivity was low and the majority of prisoners with cirrhosis were not diagnosed. As such, these algorithms perform poorly when used to diagnose cirrhosis within custodial settings and offer little clinical utility in prison HCV assessment pathways.

There is now increased focus on how best to utilise public health platforms to achieve the WHO elimination goals by 2030. Mathematical modelling has demonstrated to achieve this agenda, PWIDs must be prioritised as they contribute most to incident HCV infection [4]. The effectiveness of prison-based HCV care has now been widely described [8–10]. However, the propagation of similar programs worldwide requires simplification of HCV assessment algorithms to reduce cost, particularly for other countries operating within tight fiscal budgets. Our analysis demonstrates that APRI and FIB-4 maintain a high NPV for cirrhosis when used at their validated thresholds in the prison setting. Our new algorithm incorporating age as a categorical variable and only performing non-invasive blood test algorithms to exclude cirrhosis in those >35 years exclusively is a practical way to more aggressively reduce the number requiring LSM while maintaining a high NPV. Such strategies will expedite HCV assessment and minimise costs and therefore have an important role in the prison settings in the era of viral hepatitis elimination.

## Supporting information

**S1 Table. Accuracy of FIB4 in predicting significant fibrosis as compared to transient elastography.**
(DOCX)

**S2 Table. Performance of novel fibrosis assessment pathway applied to a validation cohort.** (DOCX)

## Author Contributions

**Conceptualization:** Timothy Papaluca, Joseph S. Doyle, Margaret Hellard, Paul Desmond, David Iser, Alexander J. Thompson.

**Data curation:** Timothy Papaluca, Anne Craigie, Lucy McDonald, Amy Edwards, Michael MacIsaac, Matthew Jarman, Tanya Lee, Hannah Huang, Andrew Chan, Mark Lai.

**Formal analysis:** Timothy Papaluca, Vijaya Sundararajan.

**Investigation:** Timothy Papaluca, Lucy McDonald, Amy Edwards, Michael MacIsaac, Jacinta A. Holmes, Matthew Jarman, Tanya Lee, Hannah Huang, Andrew Chan, Mark Lai, Vijaya Sundararajan, Jessica Howell.

**Methodology:** Timothy Papaluca, Joseph S. Doyle, Margaret Hellard, Mark Stoove, Paul Desmond, David Iser, Alexander J. Thompson.

**Project administration:** Timothy Papaluca.

**Resources:** Timothy Papaluca, Alexander J. Thompson.

**Supervision:** Jacinta A. Holmes, Joseph S. Doyle, Margaret Hellard, Mark Stoove, Jessica Howell, Paul Desmond, David Iser, Alexander J. Thompson.

**Validation:** Timothy Papaluca.

**Writing – original draft:** Timothy Papaluca, Amy Edwards, Alexander J. Thompson.

**Writing – review & editing:** Timothy Papaluca, Anne Craigie, Lucy McDonald, Michael MacIsaac, Jacinta A. Holmes, Matthew Jarman, Tanya Lee, Hannah Huang, Andrew Chan, Mark Lai, Vijaya Sundararajan, Joseph S. Doyle, Margaret Hellard, Mark Stoove, Jessica Howell, Paul Desmond, David Iser, Alexander J. Thompson.

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
