## [Decision Letter · Decision Letter 0]

27 Aug 2020

PONE-D-20-24228

Non-invasive fibrosis algorithms are clinically useful for excluding cirrhosis in prisoners living with hepatitis C

PLOS ONE

Dear Dr. Thompson,

Thank you for submitting your manuscript to PLOS ONE. After careful consideration, we feel that it has merit but does not fully meet PLOS ONE’s publication criteria as it currently stands. Therefore, we invite you to submit a revised version of the manuscript that addresses the points raised during the review process.

We look forward to receiving your revised manuscript.

Kind regards,

Chen-Hua Liu

Academic Editor

PLOS ONE

2.We note that you have indicated that data from this study are available upon request. PLOS only allows data to be available upon request if there are legal or ethical restrictions on sharing data publicly. For information on unacceptable data access restrictions, please see http://journals.plos.org/plosone/s/data-availability#loc-unacceptable-data-access-restrictions.

Reviewers' comments:

Reviewer's Responses to Questions

**Comments to the Author**

1. Is the manuscript technically sound, and do the data support the conclusions?

Reviewer #1: Partly

Reviewer #2: Yes

Reviewer #3: Yes

2. Has the statistical analysis been performed appropriately and rigorously? 

Reviewer #1: No

Reviewer #2: Yes

Reviewer #3: Yes

3. Have the authors made all data underlying the findings in their manuscript fully available?

Reviewer #1: Yes

Reviewer #2: Yes

Reviewer #3: Yes

4. Is the manuscript presented in an intelligible fashion and written in standard English?

Reviewer #1: Yes

Reviewer #2: Yes

Reviewer #3: Yes

5. Review Comments to the Author

Reviewer #1: Papaluca et al investigated the performance of APRI and FIB-4 for excluding cirrhosis in a large cohort of 1007 prisoners with hepatitis C virus infection using a liver stiffness measurement (LSM) of ≥12.5 kPa through transient elastography (TE, e.g. FibroScan) as the reference standard for cirrhosis. They demonstrated that an APRI cutoff of 1.0 and FIB-4 cutoff of 1.45 had negative predictive values (NPVs) of 96.1% and 96.6% for cirrhosis, respectively. The implementation of this triage strategy would spare the need for TE by 71%. In those aged >35, an APRI cutoff of 1.0 had a NPV of 95.1%. They concluded that such strategy can simplify the cascade of care in prisoners with hepatitis C. While the finding of this study is of interest, there are several issues that need to be addressed to improve the scientific merit of this manuscript.

Major Comments:

1. The diagnosis of cirrhosis may be relevant to the decision of DAA regimen ± ribavirin/duration and referral for the ultrasonographic surveillance of hepatocellular carcinoma (HCC) and endoscopic surveillance of gastroesophageal varices. In the setting of prisons, I am not sure how feasible is the immediate referral for surveillance of cirrhosis-related complications. Moreover, the identification of patients with advanced cirrhosis (METAVIR F3 and F4) is equally important because this subgroup of patients are at risk of HCC and therefore are recommended to continue their HCC surveillance despite achieving sustained virological response to DAA treatment. In the present cohort, there were 87 patients who were judged to have advanced fibrosis (METAVIR F3) according to LSM, which was a significant number of patients, compared with the number of patients with presumed cirrhosis (n = 124). The cutoffs of 1.45 and 3.25 for FIB-4 were originally selected to specifically rule out and maximize the positive predictive value (PPV) for diagnosing advanced fibrosis (METAVIR F3), respectively, in patients with HCV/HIV coinfection (Reference 19) and HCV monoinfection (Vallet-Pichard A et al, Hepatology 2007; 46: 32-36). Therefore, I would suggest the authors use these cutoffs to determine the sensitivity, specificity, PPV and NPV of FIB-4 for diagnosing advanced fibrosis (METAVIR F3) in the present cohort, using an LSM of ≥9.5 kPa as the reference standard. This analysis may provide us a clue about how feasible is this strategy in reducing the need for TE in identifying patients with advanced fibrosis. Perhaps by combining the use of FIB-4 and APRI at cutoffs of 1.45 and 1.0, respectively, an effective and cost-saving algorithm can be proposed to identify hepatitis C patients with METAVIR F3 and F4 for improving the cascade of care in prisoners. Please also discuss this finding.

2. The authors did not follow the standard way of presentation in describing the performance of APRI, FIB-4 and age for predicting cirrhosis in Tables 3 and 4, and Supplementary Tables 1 and 2. Please refer to the Table 4 in Reference 18 (Wai CT et al, Hepatology 2003; 38: 518-526) and Table 5A in the paper by Vallet-Pichard A et al (Hepatology 2007; 46: 32-36) for a correct presentation to allow readers to keep track of the number of patients in each category.

3. The paper by Vallet-Pichard A et al (Hepatology 2007; 46: 32-36) is more relevant to patients with HCV monoinfection. Please cite this paper in addition to Reference 19.

Reviewer #2: This study describes the utility of APRI and FIB4 in categorizing liver fibrosis severity of 1007 prisoners with chronic HCV infection. Furthermore, the utility of age threshold as a trigger for transient elastography was also explored. They found that APRI and FIB-4 scores can exclude cirrhosis in prisoners. In addition, fibroscan could be reserved for those prisoners >35 years who have APRI >1.0. The data and analysis are generally well described, and the analysis and conclusions are reasonable. This article comes to a similar conclusion about the utility of APRI and FIB-4 in hospital cohort. The major limitations include the lack of validation cohort and the operator variability in transient elastography for liver fibrosis assessment.

Specific comments

1. In the study design, the authors need a validation cohort to confirm the utility of the combination of age and APRI for liver cirrhosis assessment.

2. As a retrospective study, APRI and FIB-4 were calculated based on the laboratory parameters from chart review. Although fibrosis staging should be rather static, biochemical parameters can change more rapidly. Thus, it is important to report the mean interval between the blood tests and liver stiffness measurement.

3. The issue of interoperator variability in transient elastography for liver fibrosis may be addressed.

4. Is there any reason why the authors choose APRI, not FIB4, for fibrosis assessment in the figure 2?

5. The causes of the discordance between APRI/FIB4 and liver stiffness measurement by fibroscan should be discussed.

Reviewer #3: General comments

The study by Papaluca et al evaluated the usefulness of APRI and FIB-4 as simple markers in detecting hepatic cirrhosis in a cohort of prisoners with chronic hepatitis C. Using transient elastography (TE) as a reference, this retrospective study enrolled 1,007 patients including 124 (12%) cirrhotic patients. With cutoff 1.0 and 1.45 for APRI and FIB-4, the negative predictive value (NPV) were >96% in cirrhosis diagnosis. With the cutoff of 35 years for age, the NPV was also 96.5%. The authors proposed a simple fibrosis assessment algorithm to reduce the need of TE by 78%. For this special group of chronic hepatitis C patients, this study provided a simple and useful algorithm for cirrhosis evaluation to minimize the use of TE and barrier of HCV treatment in clinical practice. However, some points needed to be clarified.

Major comments

1. For HCV treatment with current DAA, the diagnosis of compensated cirrhosis seemed not so important in general. The authors might mention the rationale to diagnose cirrhosis for this special group of prisoners?

2. Active or previous alcohol consumption might be a problem for this special group of prisoners. Using 12.5 kPa by TE as cutoff in cirrhosis diagnosis might overestimate the patient numbers. How many patients with alcohol in this group of patients?

6. PLOS authors have the option to publish the peer review history of their article (what does this mean?). If published, this will include your full peer review and any attached files.

Reviewer #1: No

Reviewer #2: No

Reviewer #3: No

---

## [Author Response · Author response to Decision Letter 0]

13 Oct 2020

Reviewer #1: Papaluca et al investigated the performance of APRI and FIB-4 for excluding cirrhosis in a large cohort of 1007 prisoners with hepatitis C virus infection using a liver stiffness measurement (LSM) of ≥12.5 kPa through transient elastography (TE, e.g. FibroScan) as the reference standard for cirrhosis. They demonstrated that an APRI cutoff of 1.0 and FIB-4 cutoff of 1.45 had negative predictive values (NPVs) of 96.1% and 96.6% for cirrhosis, respectively. The implementation of this triage strategy would spare the need for TE by 71%. In those aged >35, an APRI cutoff of 1.0 had a NPV of 95.1%. They concluded that such strategy can simplify the cascade of care in prisoners with hepatitis C. While the finding of this study is of interest, there are several issues that need to be addressed to improve the scientific merit of this manuscript.

Major Comments:

1. The diagnosis of cirrhosis may be relevant to the decision of DAA regimen ± ribavirin/duration and referral for the ultrasonographic surveillance of hepatocellular carcinoma (HCC) and endoscopic surveillance of gastroesophageal varices. In the setting of prisons, I am not sure how feasible is the immediate referral for surveillance of cirrhosis-related complications. Moreover, the identification of patients with advanced cirrhosis (METAVIR F3 and F4) is equally important because this subgroup of patients are at risk of HCC and therefore are recommended to continue their HCC surveillance despite achieving sustained virological response to DAA treatment. In the present cohort, there were 87 patients who were judged to have advanced fibrosis (METAVIR F3) according to LSM, which was a significant number of patients, compared with the number of patients with presumed cirrhosis (n = 124). The cutoffs of 1.45 and 3.25 for FIB-4 were originally selected to specifically rule out and maximize the positive predictive value (PPV) for diagnosing advanced fibrosis (METAVIR F3), respectively, in patients with HCV/HIV coinfection (Reference 19) and HCV monoinfection (Vallet-Pichard A et al, Hepatology 2007; 46: 32-36). Therefore, I would suggest the authors use these cutoffs to determine the sensitivity, specificity, PPV and NPV of FIB-4 for diagnosing advanced fibrosis (METAVIR F3) in the present cohort, using an LSM of ≥9.5 kPa as the reference standard. This analysis may provide us a clue about how feasible is this strategy in reducing the need for TE in identifying patients with advanced fibrosis. Perhaps by combining the use of FIB-4 and APRI at cutoffs of 1.45 and 1.0, respectively, an effective and cost-saving algorithm can be proposed to identify hepatitis C patients with METAVIR F3 and F4 for improving the cascade of care in prisoners. Please also discuss this finding.

- Thank you for your suggestions.

- We are fortunate to be able to enrol our cirrhotic prisoners in hepatocellular carcinoma (HCC) and variceal surveillance whilst they remain incarcerated, and therefore it is important to identify those with cirrhosis at baseline. While alternate prison-based HCV programs may be unable to provide this surveillance, pre-treatment fibrosis determination also allows those identified as cirrhotic an opportunity to be linked to community-based surveillance following release.

- The cost effectiveness of HCC screening among patients with F3 fibrosis (LSM 9.5-12.4kPa) after effective DAA therapy remains controversial and is not current practice in Australia. This is consistent with current AASLD and Australian guidelines that recommend that only those with METAVIR F4 cirrhosis be enrolled in HCC surveillance (1, 2).

- Therefore, whilst the FIB4 cut-offs of 1.45 and 3.25 were originally validated for the diagnosis or exclusion of advanced fibrosis (METAVIR F3/F4), we specifically evaluated the performance of these cut-offs for cirrhosis (F4), as a diagnosis of advanced fibrosis (METAVIR F3 equivalent) would not influence clinical management of the prisoners. The primary focus of the analysis was evaluation of the lower threshold for excluding cirrhosis and we believe that the analysis showing that the threshold of 1.45 has a high NPV for cirrhosis is clinically useful. We included the analysis of the 3.25 threshold as it is widely published in the literature, but consistent with previous studies, it is less accurate for “ruling in” advanced fibrosis or cirrhosis

- However, we agree with the reviewer that the performance of FIB-4 for ruling in /out advanced fibrosis (LSM > 9.5 kPa) is is a relevant question that the field will be interested in, and we therefore have included two analysis for FIB-4 in the revised manuscript. This includes the accuracy of FIB-4 at thresholds of 1.45 and 3.25 for the diagnosis or exclusion of both advanced fibrosis (LSM �9.5kPa) and cirrhosis (LSM �12.5kPa)

- In our initial analysis we attempting to enhance the performance of the current fibrosis algorithms via the introduction of other variables. Ultimately, however, we identified greater utility in the incorporation of age thresholds in fibrosis assessment pathways, which were associated with high NPVs for cirrhosis, whilst more greatly reducing the need for transient elastography. 

2. The authors did not follow the standard way of presentation in describing the performance of APRI, FIB-4 and age for predicting cirrhosis in Tables 3 and 4, and Supplementary Tables 1 and 2. Please refer to the Table 4 in Reference 18 (Wai CT et al, Hepatology 2003; 38: 518-526) and Table 5A in the paper by Vallet-Pichard A et al (Hepatology 2007; 46: 32-36) for a correct presentation to allow readers to keep track of the number of patients in each category.

- Thank you for your suggestion regarding the presentation of the data. 

- Tables 3 and 4 have been updated to reflect the suggested style and have been included in the revised manuscript. As this style more accurately describes the data, including the miscategorisation of prisoners at different APRI or FIB-4 thresholds, we have removed Supplemental Tables 1 and 2.

3. The paper by Vallet-Pichard A et al (Hepatology 2007; 46: 32-36) is more relevant to patients with HCV monoinfection. Please cite this paper in addition to Reference 19.

 - Thank you for your suggestion. This has now been updated.

Reviewer #2: This study describes the utility of APRI and FIB4 in categorizing liver fibrosis severity of 1007 prisoners with chronic HCV infection. Furthermore, the utility of age threshold as a trigger for transient elastography was also explored. They found that APRI and FIB-4 scores can exclude cirrhosis in prisoners. In addition, fibroscan could be reserved for those prisoners >35 years who have APRI >1.0. The data and analysis are generally well described, and the analysis and conclusions are reasonable. This article comes to a similar conclusion about the utility of APRI and FIB-4 in hospital cohort. The major limitations include the lack of validation cohort and the operator variability in transient elastography for liver fibrosis assessment.

Specific comments

1. In the study design, the authors need a validation cohort to confirm the utility of the combination of age and APRI for liver cirrhosis assessment.

- Thank you for your suggestion to validate our novel fibrosis assessment pathway, which incorporates age and APRI thresholds. This pathway was determined by analysing our original cohort of 1007 prisoners as a practical way of minimising barriers to prison-based HCV care, and achieved a NPV of >95% for cirrhosis and minimised referral for transient elastography by 78%. 

- Since our original analysis, an additional 189 prisoners have been assessed and their data was available to contribute to a validation cohort. The prevalence of cirrhosis in this validation cohort was 11% (n=20/189). When our novel algorithm was applied to these 189 prisoners, the NPV for cirrhosis was maintained at 97% and the need for transient elastography was reduced by 83%, if only those prisoners aged 36 years or greater and who had an APRI of �1.0 were referred for this investigation.

- These results have been included in our revised manuscript.

2. As a retrospective study, APRI and FIB-4 were calculated based on the laboratory parameters from chart review. Although fibrosis staging should be rather static, biochemical parameters can change more rapidly. Thus, it is important to report the mean interval between the blood tests and liver stiffness measurement.

- Whilst minimal changes in blood-based parameters are anticipated for people living with HCV in the short- or intermediate term, our analysis excluded prisoners with an interval of greater than 6 months between bloods and TE to ensure that these results were comparable. 

- The median timeframe between bloods and transient elastography was well within this timeframe at 34 days [IQR 13-86 days], over which duration minimal changes in biochemical parameters are anticipated.

- The manuscript has been updated to include this data. 

3. The issue of interoperator variability in transient elastography for liver fibrosis may be addressed.

- Interobserver variability was identified as a potential limitation of TE during the validation of this technology. The impact of interobserver variability was minimised in this study in two key ways. Firstly, all TE examinations for prisoners included in this study were completed by one of only three hepatology clinical nurse consultants. Each nurse had undergone extensive training in TE and had performed over 100 supervised examinations. Secondly, only prisoners who had a LSM IQR/median ratio less than 0.3 were included in the analysis to minimise the inclusion of inaccurate results. 

- The manuscript has been updated to include the clinical nurse consultants TE training requirements.

4. Is there any reason why the authors choose APRI, not FIB4, for fibrosis assessment in the figure 2?

- As our proposed fibrosis assessment pathway for prisoners displayed in figure 2 includes categorical age thresholds, the APRI was selected as FIB-4 already includes age as a variable in its calculation. 

5. The causes of the discordance between APRI/FIB4 and liver stiffness measurement by fibroscan should be discussed.

- Thank you for your question.

- In all cases of cirrhosis (diagnosed by TE) where there was discordance between fibrosis algorithms and LSM, the prisoners had well compensated Child-Turcotte-Pugh class A cirrhosis, minimal transaminitis and a preserved platelet counts. For these prisoners, the most significant consideration is the missed opportunity for enrolment in HCC screening.

- The risk of HCC for these prisoners, however, will be significantly reduced with successful viral eradication (3). Therefore, the reduction in treatment throughput if TE is required for all must be considered against the reduction in liver related morbidity and mortality more broadly if processes which support prisoners being treated in large numbers are supported.

- As such, the benefit of using fibrosis algorithms in the prison is significant as they both identify the majority of cirrhotics while minimising referral for TE, improving treatment throughput. 

- We have updated the manuscript to include this consideration. 

Reviewer #3: General comments

The study by Papaluca et al evaluated the usefulness of APRI and FIB-4 as simple markers in detecting hepatic cirrhosis in a cohort of prisoners with chronic hepatitis C. Using transient elastography (TE) as a reference, this retrospective study enrolled 1,007 patients including 124 (12%) cirrhotic patients. With cutoff 1.0 and 1.45 for APRI and FIB-4, the negative predictive value (NPV) were >96% in cirrhosis diagnosis. With the cutoff of 35 years for age, the NPV was also 96.5%. The authors proposed a simple fibrosis assessment algorithm to reduce the need of TE by 78%. For this special group of chronic hepatitis C patients, this study provided a simple and useful algorithm for cirrhosis evaluation to minimize the use of TE and barrier of HCV treatment in clinical practice. However, some points needed to be clarified.

Major comments

1. For HCV treatment with current DAA, the diagnosis of compensated cirrhosis seemed not so important in general. The authors might mention the rationale to diagnose cirrhosis for this special group of prisoners?

- Thank you for your question. 

- Progression to cirrhosis in those with HCV infection is associated a risk of developing HCC and other complications of cirrhosis. Prisoners identified as cirrhotic within the Victorian prison system as enrolled in 6 monthly HCC surveillance, and where appropriate, variceal surveillance. It is therefore important to identify these prisoners prior to DAA treatment.

- Whilst the risk of HCC is reduced amongst those who achieve SVR12 (4), Australian guidelines recommend that those with cirrhosis are retained in HCC surveillance. 

- The diagnosis of cirrhosis may also impact DAA selection as EASL guidelines no longer recommend sofosbuvir/velpatasvir for first-line treatment in those with cirrhosis and GT3 infection due to poorer SVR12 rates in this group (4).

2. Active or previous alcohol consumption might be a problem for this special group of prisoners. Using 12.5 kPa by TE as cutoff in cirrhosis diagnosis might overestimate the patient numbers. How many patients with alcohol in this group of patients?

- Thank you for your question regarding comorbid alcohol misuse in this group.

- The degree of prior alcohol intake in this cohort is highly varied and challenging to accurately capture during our prison-based assessments.

- We have considered however if recent alcohol misuse could impact on TE and the biochemical indices included in FIB-4 and APRI. Further analysis of our data identified a median time of 105 days [IQR 46-272 days] between incarceration and collection of blood-based investigations. As such, as prisoners had no access to alcohol for 3 months on average, the impact of alcohol related steatohepatitis on their liver biochemistry or TE results are anticipated to be minimal.

References:

1. Marrero JA, Kulik LM, Sirlin CB, Zhu AX, Finn RS, Abecassis MM, et al. Diagnosis, staging, and management of hepatocellular carcinoma: 2018 practice guidance by the American Association for the Study of Liver Diseases. Hepatology. 2018;68(2):723-50.

2. Hepatitis C Virus Infection Consensus Statement Working Group. Australian recommendations for the management of hepatitis C virus infection: a consensus statement (June 2020). Melbourne: Gastroenterological Society of Australia, 2020.

3. Ioannou GN, Green PK, Berry K. HCV eradication induced by direct-acting antiviral agents reduces the risk of hepatocellular carcinoma. J Hepatol. 2018;68(1):25-32.

4. European Association for the study of Liver. EASL recommendations on treatment of hepatitis C 2018. J Hepatol. 2018;69(2):461-511.

---

## [Decision Letter · Decision Letter 1]

27 Oct 2020

Non-invasive fibrosis algorithms are clinically useful for excluding cirrhosis in prisoners living with hepatitis C

PONE-D-20-24228R1

Dear Dr. Thompson,

We’re pleased to inform you that your manuscript has been judged scientifically suitable for publication and will be formally accepted for publication once it meets all outstanding technical requirements.

Kind regards,

Chen-Hua Liu

Academic Editor

PLOS ONE

Reviewers' comments:

Reviewer's Responses to Questions

**Comments to the Author**

1. If the authors have adequately addressed your comments raised in a previous round of review and you feel that this manuscript is now acceptable for publication, you may indicate that here to bypass the “Comments to the Author” section, enter your conflict of interest statement in the “Confidential to Editor” section, and submit your "Accept" recommendation.

Reviewer #1: All comments have been addressed

Reviewer #2: All comments have been addressed

Reviewer #3: All comments have been addressed

2. Is the manuscript technically sound, and do the data support the conclusions?

Reviewer #1: Yes

Reviewer #2: Yes

Reviewer #3: Yes

3. Has the statistical analysis been performed appropriately and rigorously? 

Reviewer #1: Yes

Reviewer #2: Yes

Reviewer #3: Yes

4. Have the authors made all data underlying the findings in their manuscript fully available?

Reviewer #1: Yes

Reviewer #2: Yes

Reviewer #3: Yes

5. Is the manuscript presented in an intelligible fashion and written in standard English?

Reviewer #1: Yes

Reviewer #2: Yes

Reviewer #3: Yes

6. Review Comments to the Author

Reviewer #1: (No Response)

Reviewer #2: This revised manuscript is much improved and all previous comments were responded on point-to-point basis. I have no additional comments.

Reviewer #3: (No Response)

7. PLOS authors have the option to publish the peer review history of their article (what does this mean?). If published, this will include your full peer review and any attached files.

Reviewer #1: No

Reviewer #2: No

Reviewer #3: No

---

## [Editor Report · Acceptance letter]

3 Nov 2020

PONE-D-20-24228R1 

Non-invasive fibrosis algorithms are clinically useful for excluding cirrhosis in prisoners living with hepatitis C 

Dear Dr. Thompson:

I'm pleased to inform you that your manuscript has been deemed suitable for publication in PLOS ONE. Congratulations! Your manuscript is now with our production department. 

Kind regards, 

on behalf of

Dr. Chen-Hua Liu 

Academic Editor

PLOS ONE